# Plant structural diversity alters sediment retention on and underneath herbaceous vegetation in a flume experiment

Lena Kretz[1,2]*, Katinka Koll[3], Carolin Seele-Dilbat[1,2], Fons van der Plas[1,4], Alexandra Weigelt[1,5], Christian Wirth[1,5,6]

1 Life science, Systematic Botany and Functional Biodiversity, Leipzig University, Leipzig, Germany, 2 Department Conservation Biology, Helmholtz Centre for Environmental Research (UFZ), Leipzig, Germany, 3 Leichtweiß-Institute for Hydraulic Engineering and Water Resources, Technische Universität Braunschweig, Braunschweig, Germany, 4 Plant Ecology and Nature Conservation, Wageningen University, Wageningen, The Netherlands, 5 German Centre for Integrative Biodiversity Research (iDiv) Halle-Jena-Leipzig, Leipzig, Germany, 6 Max Planck Institute for Biogeochemistry, Jena, Germany

* lena.kretz@uni-leipzig.de

**Data Availability Statement:** Data are available from the Data Repository iData (https://doi.org/10.25829/idiv.3473-4-2208).

## Abstract

Sediment retention is a key ecosystem function provided by floodplains to filter sediments and nutrients from the river water during floods. Floodplain vegetation is an important driver of fine sediment retention. We aim to understand which structural properties of the vegetation are most important for capturing sediments. In a hydraulic flume experiment, we investigated this by disentangling sedimentation on and underneath 96 vegetation patches (40 cm x 60 cm). We planted two grass and two herb species in each patch and conducted a full-factorial manipulation of 1) vegetation density, 2) vegetation height, 3) structural diversity (small-tall vs tall-tall species combinations) and 4) leaf pubescence (based on trait information). We inundated the vegetation patches for 21 h in a flume with silt- and clay-rich water and subsequently measured the amount of accumulated sediment on the vegetation and on a fleece as ground underneath it. We quantified the sediment by washing it off the biomass and off the fleece, drying the sediment and weighting it. Our results showed that all manipulated vegetation properties combined (vegetation density and height, and the interaction of structural diversity and leaf pubescence) explained sedimentation on the vegetation (total $R^2 = 0.34$). The sedimentation underneath the vegetation was explained by the structural diversity and the leaf pubescence (total $R^2 = 0.11$). We further found that vegetation biomass positively affected the sedimentation on and underneath the vegetation. These findings are crucial for floodplain management strategies with the aim to increase sediment retention. Based on our findings, we can identify management strategies and target plant communities that are able to maximize a floodplain's ability to capture sediments.

## Introduction

Worldwide, intensification of agriculture leads to increasing sediment and nutrient loads in rivers [1–4]. Severe soil erosion in monocultures and excessive fertilization are main drivers of

**Funding:** C.W. received funding from the German Federal Ministry of Education and Research (BMBF) and the German Federal Agency for Nature Conservation (BfN) for the project: Wilde Mulde - Revitalization of a wild river landscape in Central Germany (Funding label: 01LC1322E).

**Competing interests:** The authors have declared that no competing interests exist.

these increased sediment and nutrient loads [5–9]. Furthermore, riparian deforestation and urbanization (through increased surface water run-off on sealed surfaces) of floodplains increase erosion, which then increases sediment and nutrient loads in the rivers [10–12]. In more natural river systems, sediment and nutrients are transported into the floodplain during flood events, where they are largely retained [13–16]. The floodplain thus acts as a sink for sediment and nutrients. However, human activities degrade river systems by straightening and embanking and thus reducing floodplain areas and their connectivity to the river [13,17]. Today, floodplains are among the most threatened ecosystems worldwide [17,18]. Consequently, river water with a higher load of sediments and nutrients needs to be filtered by strongly reduced floodplain areas [18]. The consequence is an unbalanced sediment and nutrient transport along the river which causes overfertilization of the water and silting up of sediments in river branches, oxbows and river mouths [19–23]. To improve the capacity of floodplains to retain fine sediment, and bound to it nutrients, it is crucial to obtain a holistic picture of the mechanisms optimizing fine sediment retention on natural floodplains [24]. Sedimentation can be described as a complex mixture of different biogeomorphological processes, where structural properties, such as the density or height of the floodplain vegetation, are important for deposition of fine sedimentation and the morphology of the floodplain is crucial for deposition of coarse sediment [24,25]. However, we still do not fully understand which structural properties of the vegetation are responsible for the fine sediment retention within a vegetation patch.

Different types of plant communities occurring in floodplains are known to differ in their retention capacity of fine sediment. For example, herbaceous communities differ from shrubs and tree communities [26,27], and agricultural grasslands differ from reed beds and woodlands [28]. In addition, broad-leaved and aquatic vegetation in tidal communities have a higher sediment retention capacity compared with graminoids and shrubs [29]. However, stiff grasses are known to increase sedimentation due to their high resistance to the flow [30], which reduces the flow velocity and gives the sediment time to settle [31,32]. In general, vegetation influences fluvial processes and sediment transport [33] and has the potential to double the sediment retention through its three dimensional structure [34,35]. However, while we do understand that contrasting plant functional groups (e.g. herbs versus trees) [36,37] affect overall sediment retention differently, we still have a very poor understanding of the main structural properties underlying vegetation control of sedimentation. Even within more homogeneous vegetation types, such as floodplain meadows, there is a wide variation of structural properties due to natural inter- and intraspecific variation, species sorting or grassland management. Here, we aim to understand how multiple and interacting structural properties of herbaceous plant communities drive sediment retention within vegetation patches. For the first time, we experimentally disentangle sedimentation on and underneath the vegetation, since we expect different mechanisms and thus varying importance of drivers.

The density of vegetation influences sedimentation in two ways. On one hand, dense vegetation can reduce the flow velocity, thus increasing potential time for settlement [28,38]. Further, a high density increases the standing biomass in grasslands [39] and more biomass is in turn likely to increase sedimentation, due to stronger reduction in flow velocity. On the other hand, increasing vegetation density can cause vertical mixing of the flow and increases turbulence that may change sedimentation patterns [40–42], due to remobilization of previously deposited sediment. In addition, very dense vegetation can decrease sedimentation by diverting the water flow away from the interior canopy volume, which is known as the blockage factor from studies on in-stream vegetation [32,41]. Thus, there might be a threshold of vegetation density below which sedimentation increases with density, while sedimentation again decreases once the vegetation is too dense [24]. This might be a reason why the evidence

for an effect of vegetation density on sedimentation is inconclusive so far. No density relationship at all was observed so far for tidal vegetation [29] and grass stripes [43].

Vegetation height may also affect sedimentation, since tall species probably reduce the flow velocity on a larger vertical axis than small species, which then cause sedimentation. From grassland studies it is well established that higher vegetation also has more biomass [39,44,45] and more biomass is known to capture more sediment [34,46]. However, no direct evidence was found that higher vegetation increases sedimentation in riparian buffer stripes [47].

Structural diversity, here defined as a mixture of various structural parameters of the vegetation, in this case height (small and tall growing species), can also affect sedimentation. It is typically brought forth by a high species richness [48]. Structural diversity has been found to affect the flow around in-stream vegetation, which is likely to increase sedimentation [31,49,50]. However, another experiment found that structurally more diverse vegetation patches (e.g. mixtures of structurally different plant species) did not increase sedimentation significantly compared to monocultures [51].

Sediment settles on the surface of the vegetation, thus specifically the structure of the leaf surface is important for sedimentation [34]. In a recent study, it was found that leaf pubescence explains much of the variance of sedimentation on leaf surfaces, species with hairy leaves collect more sediment than leaves of species with few or no hairs [52]. However, we expect that leaf surface structure has no strong effect on sedimentation on the soil surface underneath the vegetation, since leaves only change the flow velocity and cause turbulence close to the leaf surface [53].

The aim of this study was to understand the combined effects and relative importance of structural characteristics of the vegetation on sediment retention on and underneath herbaceous vegetation. To be able to independently assess the effects of key characteristics and their interactions, we manipulated vegetation (1) density, (2) height, (3) structural diversity, and (4) leaf pubescence in a full factorial flume experiment. To our knowledge, this is the first experiment aiming to unravel the causal relationships between sediment retention and key structural characteristics of the vegetation. We tested the following hypotheses:

1. Sediment retention on and underneath the vegetation increases with vegetation density.

2. Sediment retention on and underneath the vegetation increases with vegetation height.

3. Sediment retention on and underneath the vegetation increases with structural diversity of the vegetation.

4. Vegetation with high leaf pubescence traps more sediment on the vegetation, but not underneath the vegetation.

## Material and methods

### Vegetation patches

Our experiment comprised four treatments: leaf pubescence, structural diversity, density, and height (Fig 1). The first two treatments (leaf pubescence and structural diversity) were manipulated via species combinations during seeding, the two other treatments (density and height) were manipulated manually prior to the flume experiment. For the species selection we used the criteria leaf pubescence (hairy vs not hairy) and maximal height (<70 cm = small vs ≥70 cm = tall), and selected 3 species per category (e.g. 3 tall hairy grasses, 3 tall hairy herbs, etc.) resulting in 24 species from the German flora in order to maximize the gradient of the investigated traits (S1 Table).

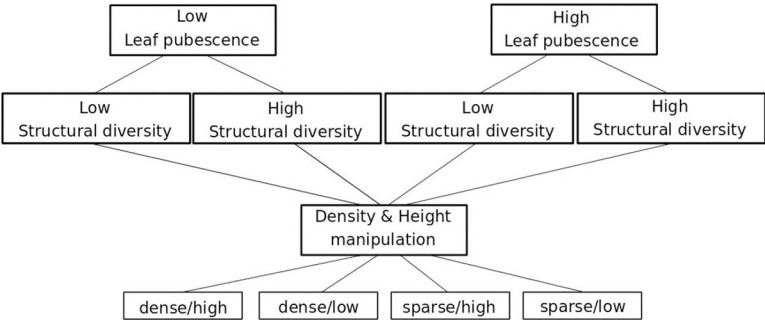

**Fig 1. Experimental design.** Low structural diversity: 2 tall grass species and 2 tall herb species, high structural diversity: 1 tall grass, 1 tall herb, 1 small grass and 1 small herb species.

For the experiment we grew 96 vegetation patches (40 x 60 cm$^2$) with 1000 seeds per patch. Always four species (2 grasses, 2 herbs; 250 seed each) were selected out of the pool of 24 species that potentially grow in Central European floodplain meadows. To manipulate leaf pubescence, half of the patches were sown with species with low leaf pubescence, and the other half with species with high leaf pubescence (Fig 1). To manipulate the structural diversity, half of the patches were sown with two tall grasses and two tall herbs (low structural diversity), while the other half were sown with one tall grass, one small grass, one tall herb, and one small herb (high structural diversity). Each treatment combination of leaf pubescence and structural diversity was replicated six times with different species combinations. Each individual species combination had four identical replicates to be able to manipulate density and height with the same species (dense/high, dense/low, sparse/high, and sparse/low, Fig 1).

Before we seeded the patches, each tray was filled with 3 cm sand mixed with fertilizer (Osmocote Exact Standard (5-6M) Meyer) as the rooting zone. On top of the sand, we put a fleece (Thermos-Fleece 85 g m$^{-2}$, Meyer). On the fleece, we spread the seeds evenly but randomly. We covered the seeds with a thin layer of a mixture of sand and turf to promote germination. The trays stood in the greenhouse of the Leipzig Botanical Garden to germinate and grow under equal conditions (Fig 2A). We seeded in June 2018 in the greenhouse without temperature regulation. We watered them once or twice per day depending on the temperature. After nine weeks in the greenhouse, we placed the trays outside, shaded by a tree to grow under natural conditions and develop higher stability and stiffness in the wind (Fig 2B). Here, they were watered automatically every 12 h by a lawn sprinkler. In total, the patches grew 13–14 weeks prior to the start of the flume experiment.

The manipulation for density and height took place in week 11, two weeks prior to the start of the experiment. For the manipulation of density and height, individuals were counted on a 2 cm x 40 cm strip in the middle of each of the four identical patches. We then used the sparser half of the patches for the sparse manipulation to keep overall density as high as possible. The density was thinned out for the "sparse" patches to a third of stems compared with the mean of the "dense" patches by manually cutting the defined number of randomly selected stems just above the fleece (S1 Fig). For the height treatment, the "high" patches were cut to 40 cm and the "low" patches to 20 cm height.

## Experimental set-up

We ran the experiment in a flume (Stahl-Technik-Straub GmbH & Co KG) in the hydraulic laboratory of the Leichtweiß-Institute for Hydraulic Engineering and Water Resources at the TU Braunschweig. The flume was 30 m long, 2 m wide and 0.8 m deep. For the experiment,

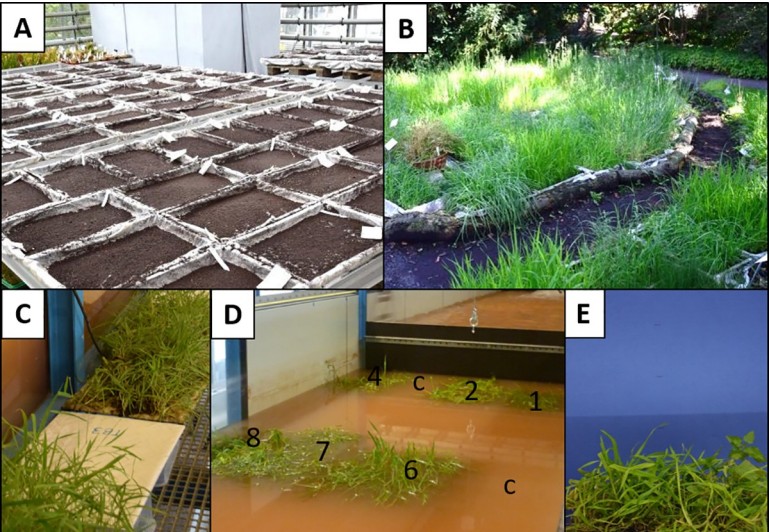

**Fig 2. Experiment preparation.** (A) seeded patches in the greenhouse, (B) patches growing outside, (C) fleeces and vegetation before the experiment, (D) patches in the flume while the water level goes down (numbers indicate patch position and c the controls), (E) image of a patch from the front.

we used a closed water cycle powered by a pump recirculating the water and the dissolved sediment with a constant discharge. Based on the results from pre-experiments, the technical limitations of the flume and the expert knowledge of hydraulic engineers, we decided for a constant discharge of 24–25 l s$^{-1}$, to be able to keep the balance between a low discharge simulating real floodplain inundation and a discharge high enough to keep the clay and silt floating in the flume water. The flume was filled up to a water depth of 45 cm (limited by the experimental set-up), so that all patches (20 cm and 40 cm manipulated height) were fully inundated. The whole waterbody of 28 000 l was mixed with the sediment by adding 7.5 kg clay (Ø <2 μm) and 7.5 kg silt (Ø 2–63 μm (90%)) before the experiment started. We decided for a mixture of clay and silt, since the small grain sizes are important for associated nutrient retention and additionally we do not expect courser sediment to settle on the vegetation due to their heavier weight. The first 5 m of the flume bed (inlet section) was covered with artificial lawn and bricks to roughen the flow bed and ensure fully turbulent flow conditions and a uniform flow field across the flume width upstream of the first line of vegetation patches (Fig 3). The vegetated flume section was separated by two closing walls and drained by a mobile submerged pump to enable removal of the patches without distorting the sediment by lifting the patches through the whole water column. With the overall constant hydraulic conditions we were able to focus on the relative differences in the structural characteristics of the vegetation during inundation.

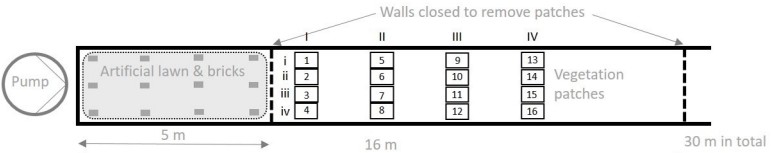

**Fig 3. Top view of flume.** Sketch of flume with the pump, the inlet section to roughen the bed (artificial lawn and bricks), the closable walls to drain the vegetated section, and the positions of the patches in lines (capital Roman numerals) and rows (lowercase Roman numerals).

## Experimental procedure

We prepared the patches for the experiment by removing the fleece with plants from the trays and washing off the sand between the roots with water. Each fleece with the aboveground parts above the fleece and the roots below was fixed on a metal plate (40 x 60 cm$^2$) with magnets (Fig 2C). The magnets were further used to keep the metal plates at their position in the flume (Fig 3). Within one run 16 metal plates were processed, where four were control patches with just blank fleeces to measure sedimentation in the absence of vegetation, and the other twelve carried vegetation patches. Each line and each row had a control patch, but the positions shifted for each run (Fig 2D). The patches per run and the position in the flume were randomly selected. In total, we conducted 9 runs, 8 of them with each 12 vegetation patches and 4 control patches and one additional run with 16 control patches.

Prior to every run we took lateral images of every patch from all four directions. The camera settings and distances were kept constant against a standard blue screen (Fig 2E). We used these images as additional variables to describe the density and the height structure of the single patches. Two to three hours after starting a run, we measured the flow velocity (except for the first run, due to technical problems) 10 cm upstream of each patch at a depth of 10 cm above the metal plate with a hydrometrical current meter (OTT C2, OTT HydroMet) twice for 30 s. We took reference water samples 2–3 h prior to the end of a run, 100 ml each on four different positions in the flume (two upstream of the plant patches and two downstream, all were taken at the water surface). At the end of a run (20–22 h) we closed the walls upstream of and downstream of the patches and slowly pumped the water out of the vegetated flume area (Figs 2 and 3). Thus, we ensured that we did not distort the samples or lose sediment during the removal of the patches from the flume. After removing the patches, we opened the walls again and the sediment settled on the ground was remobilized using a scrubber. As a last run, we did an additional control run with just 16 control fleeces as reference for sedimentation and flow measurements.

## Sample processing

The patches were processed immediately after removing them from the flume. On each patch the plants were carefully harvested (cut with a scissor just above the fleece), and washed to collect the accumulated sediment ('sediment on the vegetation' hereafter). After washing, the aboveground plant biomass was dried for at least 18 h at 100˚C and weighted. All fleeces, vegetated as well as control, were washed to collect the sediment accumulated on the ground underneath the vegetation ('sediment underneath the vegetation' hereafter). The washing water of both vegetation and fleeces was collected and stored for later processing.

The water samples were kept cool in the lab for a maximum of 5 days and later in a fridge at 4˚C for up to 3 months until all samples were processed. We filtered the sediment rich water (2 mm) to remove coarse sediment, turf and organic material, filled it into glass beakers, dried those at 110˚C and weighted the absolute amount of sediment (g) per fleece (patch of 0.24 m$^2$) and per biomass (g) per patch (patch of 0.24 m$^2$).

## Statistics

All statistical analyses were done with the statistical software R [54]. We ran two separate models to investigate which factors drove sedimentation on the vegetation and on the fleeces underneath the vegetation. In a simple linear model the flow velocity did not correlate with the position within the flume; however, the first patch line (Fig 3) had significantly higher flow velocity. For the missing flow velocity measurements of the first run the means per position lines were used to correct for flow velocity per patch. We observed a significant constant loss

of sediment over time on the control fleece and in the daily sediment sample, when we tested with a linear model including run identity, position identity and flow velocity as predictor variables ($R^2$ = 0.31). This is likely due to sedimentation within the tube that transported the water from the end of the flume to its start. To correct for this background sediment loss over time we ran two mixed effect models (lmer function, lme4 library [55]) with sedimentation on or underneath the vegetation as the response variable, the run identity (day) as fixed factor and the position in the flume as random factor. In a next set of linear models, we used the residuals of the first mixed effect models as the response variables, to explain the remaining variance in sedimentation that was not explained by the run identity or the position within the flume. As explanatory variables we used the experimental treatments (leaf pubescence, structural diversity, density, and height) and the flow velocity. We also included interaction effects that we hypothesized to be important: an interaction between density and height as well as the interaction between structural diversity and leaf pubescence. We compared our 'full model' (including all initial predictors) with simplified models, and selected the most parsimonious model based on the lowest AIC value, using the stepAIC function of the MASS library, [56]. The model residuals were checked for normality. We ran two additional linear models with the residuals of the sediment on and underneath the vegetation as response variable. But this time, instead of using the factor levels, we used continuous variables of the vegetation structure that have been measured (instead of the manipulated factors) as explanatory variables, in order to find out which structural properties of the vegetation were most influential in driving sedimentation, and to understand how these support our main results. The values for the actual structural parameters for the patches was calculated from the lateral images. We calculated four structural parameters of the vegetation: vertical density, mean height, median height, and standard deviation of the height (S2 Table). The images were colour normalised and resampled from a resolution of 4000 by 6000 pixels to a resolution of 400 by 600 pixels and afterwards transformed into grey-scale images. In order to perform a binary classification of the image into vegetation and background, we used the otsu-tresholding method [57], as implemented in the package EBImage [58]. We used the mean of all four images (front, left, back, and right) as variable in the models; additionally, we included the total plant biomass and the flow velocity in the model. The model selection procedure was the same as for the first two models. We scaled and centred all continuous variables, and removed mean height and median height as variables due to multicollinearity (variation inflation factor above 5.0, vif function, car library, [59]) and selected the best model fit.

## Results

Overall, our results showed that sediment underneath the vegetation had a mean mass of 4.83 ±1.68 g patch[-1] (ranging from 2.03 g patch[-1] to 10.62 g patch[-1]) and the sediment on the vegetation had a mean mass of 4.38±2.84 g patch[-1] (ranging from 0.74 g patch[-1] to 17.37 g patch[-1]). Mean flow velocity was 0.45±0.44 m s[-1] (ranging from 0.09 m s[-1] to 2.33 m s[-1]).

### Sedimentation on the vegetation

The sedimentation on the vegetation was explained by the vegetation density, the vegetation height and the interaction between structural diversity and leaf pubescence ($R^2$ = 0.34, Table 1). Patches that had a high structural diversity in combination with high leaf pubescence collected more sediment on the vegetation than all other patches ($p$ = 0.01, Fig 4A). Further, sediment on the vegetation was higher in patches with a high vegetation density ($p$<0.01, Fig 4B), and also in patches with a high vegetation height ($p$ = 0.01, Fig 4C).

**Table 1. Statistical model results.**

| | Residuals of sediment on the vegetation | | | | |
|---|---|---|---|---|---|
| | **Estimate** | **Std. Error** | **t value** | **Pr(>\|t\|)** | **Sig.** |
| **(Intercept)** | -2.300 | 0.539 | -4.270 | 4.84E-05 | *** |
| **Density** | 1.944 | 0.440 | 4.420 | 2.75E-05 | *** |
| **Height** | 1.156 | 0.440 | 2.629 | 0.010 | * |
| **Leaf pubescence** | 0.416 | 0.622 | 0.668 | 0.506 | |
| **Structure diversity** | -0.128 | 0.622 | -0.205 | 0.838 | |
| **Structure diversity x Leaf pubescence** | 2.423 | 0.880 | 2.755 | 0.007 | ** |
| | Residuals of sediment underneath the vegetation | | | | |
| | **Estimate** | **Std. Error** | **t value** | **Pr(>\|t\|)** | **Sig.** |
| **(Intercept)** | 0.452 | 0.155 | 2.919 | 4.40E-03 | ** |
| **Leaf pubescence** | 0.539 | 0.179 | -3.015 | 0.003 | ** |
| **Structure diversity** | 0.365 | 0.179 | -2.041 | 0.044 | * |

Sig. indicates the significance (*p<0.05, **p<0.01, ***p<0.001).

## Sedimentation underneath the vegetation

Sedimentation on the fleeces underneath the vegetation was less well explained by the leaf pubescence of the species and the structural diversity ($R^2$ = 0.11, Table 1). Sedimentation was higher in patches with higher structural diversity (*p* = 0.04, Fig 5A), and in patches with pubescent species (*p*<0.01, Fig 5B).

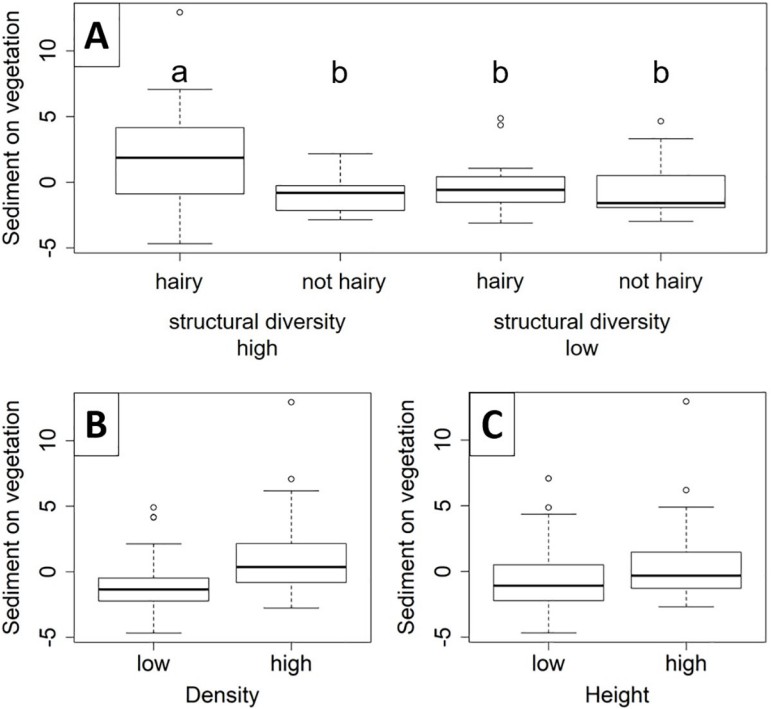

**Fig 4. Sediment on vegetation.** Residuals of sediment on vegetation explained by (A) the interaction between structural diversity and pubescence (p = 0.007, lower case letter (a and b) indicating the significant different groups), (B) density (p<0.001) and (C) height (p = 0.013).

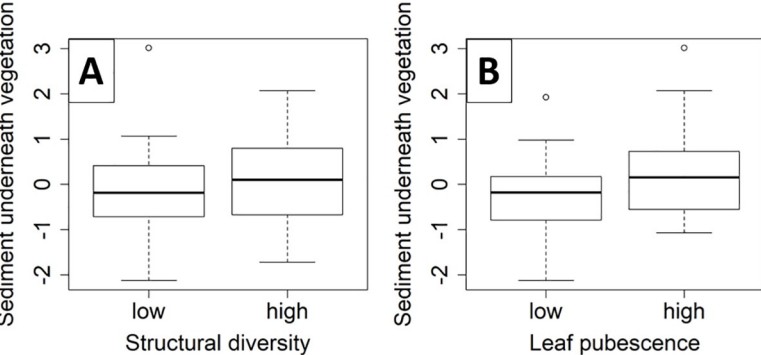

**Fig 5. Sediment underneath vegetation.** Residuals of sediment underneath vegetation explained by (A) structural diversity ($p = 0.044$) and (B) leaf pubescence ($p = 0.003$).

### Effect of measured vegetation characteristics

From the additional analysis of the measured vegetation characteristics per patch, we found that the log biomass and the height variation explained sedimentation on the vegetation ($R^2 = 0.30$, S3 Table). With an increasing amount of biomass, sediment on the vegetation increased ($p<0.01$, Fig 6A), while it decreased with increasing height variation ($p = 0.03$, Fig 6B). Sedimentation underneath the vegetation was explained by the biomass and the vertical density ($R^2 = 0.07$, S3 Table). Sedimentation increased with increasing biomass ($p = 0.01$, Fig 7A) and decreased with increasing vertical density ($p = 0.01$, Fig 7B).

## Discussion

With our experiment, we investigated the effect of mutually independent structural characteristics of the vegetation on sediment retention within vegetation patches. Our results showed that all experimentally manipulated structural characteristics (vegetation density, height, structural diversity and leaf pubescence) explained sedimentation on and underneath the vegetation. For the sedimentation on the vegetation, our data corroborates our hypotheses 1 to 4 that with a higher density, height, and the interaction of structural diversity and leaf pubescence the vegetation patches trap more sediment. Sedimentation underneath the vegetation was, as expected, significantly explained by structural diversity (hypothesis 3) and, contrary to our expectation, significantly explained by leaf pubescence (hypothesis 4). The results of the structural characteristics that we measured per patch were also mostly in line with our hypotheses.

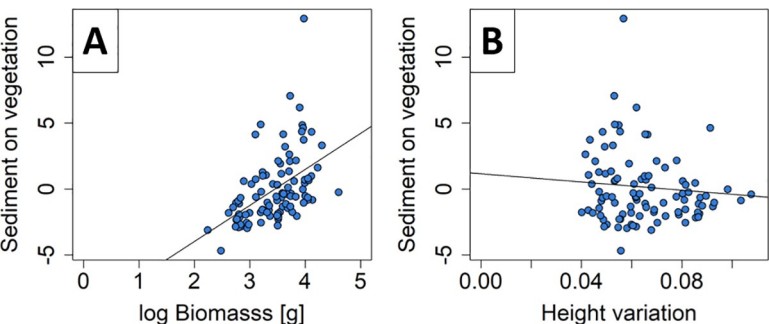

**Fig 6. Measured variables on vegetation.** Residuals of sediment on vegetation explained by (A) log biomass ($p<0.001$) and (B) height variation ($p = 0.034$).

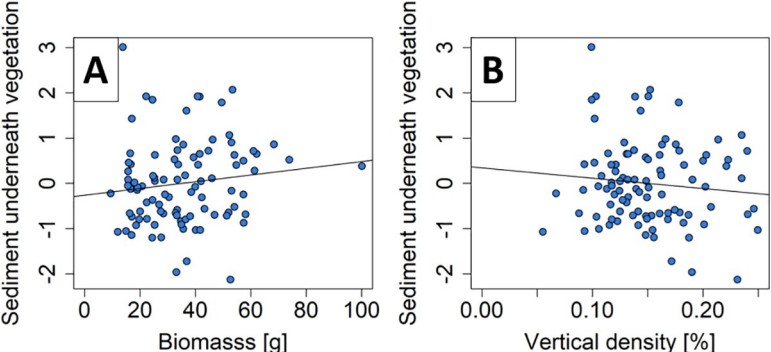

**Fig 7. Measured variables underneath vegetation.** Residuals of sediment underneath vegetation explained by (A) biomass ($p$ = 0.007) and (B) vertical density ($p$ = 0.010).

Increasing biomass increased sedimentation, which supports hypotheses 1 (density) and 2 (height). The results that more evenly tall vegetation increased sedimentation on the vegetation may support the expectation that denser patches accumulate more sediment (hypothesis 1), while underneath the vegetation the vertical density decreased sedimentation.

## Structural diversity and leaf pubescence

The structural diversity and the leaf pubescence had a significant positive effect on sedimentation. A mixture of species with tall and small stature increased sedimentation compared to just tall growing species, while species with hairy leaf surfaces additionally increased it.

Sedimentation on the vegetation was highest on patches with high structural diversity and leaf pubescence (Fig 4A). It might be that the tall (and hairy) plants reduce the flow velocity and thus enable higher sedimentation on the smaller, hairy plants below the canopy. In this case, it would be interesting to further investigate if patches with tall, non- hairy species, combined with small hairy species would have the same effect. It has been found for in-stream vegetation that structurally diverse patches increase the flow resistance [32,49], consequently reduce the flow velocity within the vegetation patch, and sediment deposits [31,32]. The probability to capture sediment is higher for species with hairy leaves, which was recently confirmed in another study [52]. It was also found in studies on airborne particles that high pubescent plant leaves collect more particles on the leaf surface [60,61].

For sedimentation underneath the vegetation, we found that more structurally diverse patches accumulate more sediment and contrary to our expectations, also leaf pubescence increased sedimentation underneath the vegetation. The combination of tall and small plants seems to reduce flow velocity more strongly close to the ground resulting in higher sedimentation rates. However, also contrasting results have been found for erosion buffer stripes, where morphologically diverse vegetation (mixture of grasses, shrubs and young trees) reduced sediment retention compared to morphologically homogeneous vegetation [62]. Surprisingly, also the leaf pubescence increased sedimentation underneath the vegetation. We speculate that this is caused by sediment that first settles on the leaf surfaces and afterwards falls down from the leaves onto the fleece underneath the vegetation. This might have happened during the time when the water level was lowered at the end of the experiment and leaf surfaces started to dry before the harvest. Alternatively, species with pubescent leaves reduce the flow velocity more strongly within the vegetation patch where the flow resistance is higher, which may again give the sediment more time to settle [28,38].

In our experiment, we manipulated the structural diversity by species selection based on their natural growth height, but we kept the number of species and individuals per patch constant and manipulated the total density and height. In a natural grassland system however, structural diversity can be caused by species diversity [48], not species identity alone. This might imply that more species rich meadows are able to capture more sediment due to their higher structural diversity.

## Vegetation structural characteristics

We found evidence that vegetation density and vegetation height significantly increased sedimentation on the vegetation, but not underneath it. We further found that some of the measured structural characteristics of the vegetation, the biomass, the vertical density and height variation were relevant for sedimentation on and underneath the vegetation.

The fact that sedimentation on the vegetation was higher on denser and taller vegetation can partly be explained by the positive effect of plant biomass on sedimentation. For grasslands it has been found that biomass correlates with height and density of the vegetation up to a specific point beyond which biomass saturates [39]. More biomass provides more surface where the sediment can settle. Denser and higher vegetation with more biomass creates stronger resistance to the water, which reduces the flow velocity within the vegetation patch [30,38,63]. This creates conditions, where sediment settles on the vegetation surface. Similar results were found for in-stream vegetation [32]. However, this study measured sediment retention downstream of a vegetation patch, and can only estimate the overall sedimentation. Our study, for the first time, distinguished the effects of sedimentation on and underneath the vegetation under experimental conditions. We additionally found that sediment on the vegetation significantly decreased with vegetation height variation, where more even canopy accumulated more sediment on the vegetation, even though the patches were cut to specific heights. We can only speculate about the mechanism causing this effect, but one reason can be that patches strongly varying in height, may experience stronger turbulences within the height depressions. Stronger turbulences then probably reduce sedimentation due to moments of high flow velocity [24]. Interestingly, the same effect was found in an observational field study (unpublished work), using the same image technique on floodplain vegetation after a flood event. For grasslands it was found that the inequality of the height distribution is positively correlated with species diversity, and negatively with productivity [64]. This suggests that species diversity decreases sedimentation via height inequality (high height variation), while at the same time increases sedimentation via a higher biomass production.

Sedimentation underneath the vegetation also increased with plant biomass and decreased with increasing vertical density of the vegetation, even though the manipulated density did not show any significant effect. There are two possible explanations: First, an interception effect of the vegetation can cause that very dense vegetation itself captures the sediment. However, that contradicts our finding that higher biomass, which also had higher density, increased sedimentation. Second, very dense vegetation may hinder or block the water to flow through the patch (blockage factor for in-stream vegetation [32]) which probably reduces sedimentation within the patch. The vertical density is an indicator for the flow resistance of the patch. More flow resistant vegetation has been found to increase sedimentation around vegetation in streams and on floodplains [32,49]. However, this may reduce the sedimentation within the patch itself, which is still contrary to our findings regarding density and biomass.

It is important to note that we manipulated vegetation height, thus the height variation and also the vertical density deviate from natural communities. However, our experiment gives important insights into causal relationships between structural vegetation characteristics and

sedimentation. Our results hint towards the potential importance of species diversity to increase sedimentation in floodplain vegetation as diversity is known to increase vegetation biomass [65], height [66], and density [67], all of which positively affect sedimentation. However, too dense patches may also reduce sedimentation by the blockage effect and increased turbulences in micro-depressions [24,32].

## Abiotic conditions

The focus of our study was on the effect of the vegetation structure on fine sediment retention during inundation. However, the abiotic conditions of the flood and the floodplain are also crucial for sedimentation on floodplains. Floods differ in their characteristics (discharge, velocity, duration, and sediment load) and floodplains differ in their topography (elevation, slope, and connectivity). The flow velocity determines, on one hand, the transport capacity of the flow and affects, on the other hand, structural parameters of the vegetation due to bending and streamlining of the plants and, thus, is a key parameter for sedimentation [49,68,69]. Moreover, the inundation depth on floodplains is an important parameter for how much sediment the vegetation is able to capture [70]. In nature, sediment load and grain size distribution of a flood water depend on the geology, the topography and the land use of the river catchment [71,72].

In our experiment, we simulated simplified a flood with constant abiotic and hydraulic conditions in order to focus on the effects of the plants characteristics. Therefore, we selected a fixed discharge, water depth and, thus, approach velocity. The variation of the flow velocity within the experiment section was measured and considered in the statistical analysis. We further focused on small grain sizes (silt and clay), which are highly relevant for nutrient retention and are likely to get trapped by the structure of the vegetation. Future studies should assess whether our main conclusions regarding the importance of different structural properties of the vegetation are also valid for differing abiotic or hydraulic conditions.

## Conclusion

With our study, we could prove that the vegetation structure plays an important role for sediment retention on floodplains. Our study is the first experiment to look inside the vegetation patch to understand sedimentation on the vegetation and directly underneath it. Our experimental design with constructed species communities gave the unique chance to unravel sediment retention caused by different, non-correlated structural characteristics of the vegetation. The relatively low values for $R^2$, however, suggest that there must be other factors not manipulated or measured in this experiment which may explain sediment retention. Other vegetation characteristics that have been identified to change the flow resistance and flow velocity around vegetation, and thus potentially sedimentation, are bendiness of the species, elasticity and rigidity of the single individuals and the stem/leaf ratio [73–75]. From our results, we can derive some management strategies for sediment retention on floodplain meadows. First, promotion of structural diversity would increase sedimentation, thus supporting species rich plant communities is likely to also increase sediment retention, with additional benefits for other ecosystem functions on floodplains. Second, higher and denser vegetation and more standing biomass, increases sedimentation. Therefore, reduced mowing in late summer would increase standing biomass during wintertime, when floods most often occur. Third, the abundance of pubescent species increases sedimentation, which could guide species selection for floodplain restoration. Overall, we can state that meadows with higher vegetation density and height, higher structural diversity and leaf pubescence increase sedimentation and with that the capacity of floodplains to fulfil the ecosystem service of sediment and nutrient retention.

## Supporting information

**S1 Fig. Stems per patch.** Mean stems per patch (40 cm x 60 cm) for the two dense patches per group (same species) and the density treatment sparse (reduced to 1/3 of the dense treatment per group). The individuals were counted on a 40 cm x 2 cm stripe in the middle of the patch. (TIF)

**S1 Table. Species selection.** Species selection of the experiment according to leaf pubescence, growth height and growth form.
(DOCX)

**S2 Table. Predictor definitions.** Definition of the predictor variables calculated from the images. *Without a unit, since length was not scaled.
(DOCX)

**S3 Table. Statistical model results.** Residuals of sediment on the vegetation and underneath the vegetation explained by the measured variables.
(DOCX)

## Acknowledgments

First, we want to thank the workgroup of the Leichtweiß-Institute for Hydraulic Engineering and River Morphology of Prof. Jochen Aberle for the possibility to run the experiment in their experimental flume. Thanks for the friendly and constructive support of experienced engineers and the associated workshop. Further, we thank Ton Hoitink from the Department of Environmental Science, subdivision Hydrology and Quantitative Water Management at the University of Wageningen (NL) for the possibility to run our pre-study in their experimental flume. We thank The Botanical Garden of Leipzig, especially Stefan Lütjens for the extensive support regarding the propagation and the plant growth. Further, we thank Ronny Richter for the support analysing the images and the student lab assistants Maria Kahler, Karl Andraczek, Georg Rieland and Paul Lyam for their great help preparing and running the experiment and processing the samples.

## Author Contributions

**Conceptualization:** Lena Kretz, Katinka Koll, Carolin Seele-Dilbat, Fons van der Plas, Alexandra Weigelt, Christian Wirth.

**Formal analysis:** Lena Kretz, Carolin Seele-Dilbat, Fons van der Plas.

**Investigation:** Lena Kretz.

**Methodology:** Lena Kretz.

**Project administration:** Christian Wirth.

**Resources:** Katinka Koll.

**Writing – original draft:** Lena Kretz.

**Writing – review & editing:** Katinka Koll, Carolin Seele-Dilbat, Fons van der Plas, Alexandra Weigelt, Christian Wirth.

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
