## [Decision Letter · Decision Letter 0]

13 Jan 2021

PONE-D-20-34689

Plant structural diversity alters sediment retention on and underneath herbaceous vegetation in a flume experiment

PLOS ONE

Dear Dr. Kretz,

Thank you for submitting your manuscript to PLOS ONE. After careful consideration, we feel that it has merit but does not fully meet PLOS ONE’s publication criteria as it currently stands. Therefore, we invite you to submit a revised version of the manuscript that addresses the points raised during the review process.

I agree with reviewers’ comments. The paper could be accepted for publication after addressing reviewers concerns. It would probably require a minor revision. A careful English edition of the main text is desirable. Overall the paper is clear and well-written, but there are several typos and sentences that could be written more clearly. Please note that once the paper is accepted, PLoS One will not send proofs to the authors, so it is important to edit the paper as if it were final.

We look forward to receiving your revised manuscript.

Kind regards,

Cristina Armas

Academic Editor

PLOS ONE

Journal Requirements:

Additional Editor Comments:

Beside from comments from reviewers, please check:

There are some “effects” that should be replaced by affect (e.g., L. 36, L. 72 and so on).

Results: Should the units of sediment mass (g) be referred to some area unit (each g m-2)?

Please check the letters included in the Figures so as not to use same letters with different purposes (i.e., results of post-hoc tests and panel “order”.

Reviewers' comments:

Reviewer's Responses to Questions

**Comments to the Author**

1. Is the manuscript technically sound, and do the data support the conclusions?

Reviewer #1: Yes

Reviewer #2: Yes

2. Has the statistical analysis been performed appropriately and rigorously? 

Reviewer #1: Yes

Reviewer #2: Yes

3. Have the authors made all data underlying the findings in their manuscript fully available?

Reviewer #1: Yes

Reviewer #2: Yes

4. Is the manuscript presented in an intelligible fashion and written in standard English?

Reviewer #1: Yes

Reviewer #2: Yes

5. Review Comments to the Author

Reviewer #1: General Comments

The article by Dr. Lena Kretz and colleagues entitled ‘Plant structural diversity alters sediment retention on and underneath herbaceous vegetation in a flume experiment’ explore the effect of aboveground vegetation structure on sediment dynamics on floodplains. This research is based on an ex situ experiment undertaken within a flume of 30m long with natural totaly submerged vegetation. The originality of the approach is the consideration of vegetation structural variably (density, height, structural diversity, and leaf pubescence) on fine sedimentation trapping (i) on the plant and (ii) underneath.

Overall the paper is clear and comprehensive. The experiment is well designed and the statistical analyses are adequate and well used. The results are interesting. They bring novelty and improve the way plant effects on sediment dynamics need to be considered in experimental approaches. In particular, they point out the necessity to quantitatively consider the structural diversity (potentially inter- and intra-specific) of plant morphological traits. The discussion is convincing and opens interesting perspectives for future experimental investigations. It would be relevant to add a paragraph in the discussion section related to the fact that the runs were conducted with the same water height. Changes in the hydrogeomorphological parameters (sediment supply and texture, channel slope, discharge) may affect vegetation effects on sediment trapping. Plant structural diversity could possibly explain sediment trapping variability only in a restricted range of discharge and sediment load. I believe you need to acknowledge that point.

Specific comments

L. 36. ‘vegetation biomass positively effects’. Replace ‘effects’ by ‘affects’

L. 44-45. ‘Thus, floodplains are among the most threatened ecosystems worldwide’ The causal linkage with fine sediment dynamics is not obvious. Clarifications are required about why floodplains are to be considered as threatened ecosystems in relation with fine sediment load.

L. 46. How does urbanization of floodplains increase erosion?

L. 71. ‘Plant functional groups’ require a definition from the classical functional trait-based approach (e.g. Garnier, Lavorel, Diaz papers).

L. 75. ‘due to natural variation’. What is the natural variation? Variation of what (genotypic, phenotypic, phenology, habitat conditions…)?

L. 85-86. ‘On the other hand, dense vegetation can cause vertical mixing of the flow and increase turbulence that may change sedimentation patterns’. A vegetation matt with intermediate density can also enhance turbulence and erosion by causing flow divergence around the structures.

L. 105-106. What here is the exact meaning of ‘functionally more diverse vegetation patches (e.g. mixtures of functionally different plant species)’. Functionality (related to sediment trapping? Or to other functions such as photosynthesis if your consider leaf area…) needs to be better defined from the beginning.

L. 112-113. ‘However, we expect that leaf surface structure has no strong effect on sedimentation on the soil surface underneath the vegetation.’ Why? You need to provide a short explanation. Large leafs could increase biomass and thus water blockage effect? But indeed leaf pubescence should not affect underneath sedimentation.

L. 336. Replace (Kretz et al. 2020) by the ref number.

L. 357. Replace (Proulx et al. 2014) by ref number.

Reviewer #2: The authors present an interesting study on the effect of vegetation characteristics on sedimentation in a flume environment. A diverse set of vegetation patches was seeded and manipulated for the experiments. The sedimentation on the plants versus “underneath the vegetation” was investigated and the effect of vegetation density, height, structural diversity, and leaf pubescence studied. The results confirmed their hypotheses and are discussed with other literature findings on the effect of in-stream vegetation on sedimentation processes.

The paper is well-written and structured. The methodology is clear, and the results are discussed in detail. It is an interesting contribution and I only have some minor comments regarding the flume experiments and test program as well as some suggestions regarding the figures. Based on my opinion, the paper only needs minor revision.

Comments per section:

Please check “affect” and “effect” in the text (L36; L72)

Abstract

1. Please add 1-2 sentences on how the sedimentation was determined (as described in Sample processing)

Introduction

2. L53-54: Add sediment and “nutrient transport”, as in L54 you refer to “overfertilization”

Experimental set-up

3. L175: Why is it important to state the company that built the flume?

4. L182: What do you mean by “roughen the flow” – that they acted as a flow straightener to suppress secondary currents and establish uniform flow conditions? Please revise.

5. L183 and in the text: Please refer to “upstream and downstream” and not “in front and behind”

6. L185: under dry conditions instead of at ?

7. It is not clear to me how the water and the sediment / clay was mixed and where. Was there a recirculating pump and the mixture of water and sediment was flowing through the test section with the respective discharge? Did the sediment settle upstream of the test section? Did you add the sediment prior to every run? It would be helpful to add some more information to the text and also a test program. It is not so clear to me how many runs were conducted.

8. Regarding the flow depth: Have you considered to perform flume experiments under non-submerged, so emergent conditions or vary the discharge? Did you choose the discharge and flow depth based on the flume capacity? Why were these parameters not varied / discussed?

Experiment conduction (I recommend Experimental procedure)

Results

9. L273: The variation of the flow velocity is similar to the mean value. Why? Was it due to the measurement device?

Discussion

10. Similar to comment 8, it would be interesting to discuss the tested flow conditions (discharge, flow depth / submergence level) and characteristics of sediment and clay particles in the flow – and how these parameters affect the sedimentation processes. The flow velocity and particle size will affect the sedimentation processes, so more information on how you chose them would be interesting – also with respect to how to upscale them to prototype conditions.

Figures

11. Fig. 3: What does a – b – b – b stand for above the box plot? It looks similar to the indication of the subfigures; please adapt

12. Fig. 5-6: I recommend starting with 0 at the x-axis

13. Fig. S1: I highly recommend including this figure in the main text, as it nicely illustrates what the experiments looked like

6. PLOS authors have the option to publish the peer review history of their article (what does this mean?). If published, this will include your full peer review and any attached files.

Reviewer #1: No

Reviewer #2: No

---

## [Author Response · Author response to Decision Letter 0]

16 Feb 2021

Responses to the reviewer and editor comments can be find in the document "Response to Reviewers".

---

## [Editor Report · Decision Letter 1]

24 Feb 2021

Plant structural diversity alters sediment retention on and underneath herbaceous vegetation in a flume experiment

PONE-D-20-34689R1

Dear Dr. Kretz,

We’re pleased to inform you that your manuscript has been judged scientifically suitable for publication and will be formally accepted for publication once it meets all outstanding technical requirements.

Kind regards,

Cristina Armas

Academic Editor

PLOS ONE
---

## [Editor Report · Acceptance letter]

9 Mar 2021

PONE-D-20-34689R1 

Plant structural diversity alters sediment retention on and underneath herbaceous vegetation in a flume experiment 

Dear Dr. Kretz:

I'm pleased to inform you that your manuscript has been deemed suitable for publication in PLOS ONE. Congratulations! Your manuscript is now with our production department. 

Kind regards, 

on behalf of

Dr. Cristina Armas 

Academic Editor

PLOS ONE